# Successful Treatment of a COVID-19 Case with Pneumonia and Renal Injury Using Tocilizumab

**Yugo Ashino [1], Haorile Chagan-Yasutan [2,3], Masumitsu Hatta [4], Yoichi Shirato [1], Yorihiko Kyogoku [1], Hanae Komuro [1] and Toshio Hattori [3,\***

1 Department of Respiratory Medicine, Sendai City Hospital, Miyagi 982-8502, Japan; ya82@yahoo.co.jp (Y.A.); youichi0520@icloud.com (Y.S.); y.kyogoku@rm.med.tohoku.ac.jp (Y.K.); H.KOMURO@RM.med.tohoku.ac.jp (H.K.)
2 Mongolian Psychosomatic Medicine Department, International Mongolian Medicine Hospital of Inner Mongolia, Hohhot 010065, China; haorile@gjmyemail.gjmyy.cn
3 Department of Health Science and Social Welfare, Kibi International University, Okayama 716-0018, Japan
4 Department of Infectious Diseases Medicine, Sendai City Hospital, Miyagi 982-8502, Japan; hattamasumitsu@yahoo.co.jp
\* Correspondence: hattorit@kiui.ac.jp; Tel.: +81-866-22-9454

**Abstract:** A 49-year-old male Japanese patient was admitted to our hospital under the diagnosis of COVID-19 pneumonia. For 5 days before admission, he had experienced various symptoms, including high fever, watery diarrhea, dyspnea, and cough, and he tested positive for severe acute respiratory syndrome coronavirus 2 (SARS-CoV-2) nucleic acid. The patient is a smoker who was on medication for hypertension. A chest computed tomography scan showed bilateral multiple patchy ground-glass opacities. Despite being treated with several therapeutic agents, he still exhibited dyspnea (oxygen saturation [$SpO_2$] in ambient air: 88%), a high fever (axillary temperature: 39 °C), and high blood pressure (148/98 mmHg). Because laboratory data revealed high levels of C-reactive protein (CRP; 2.10 mg/dL) and urinary β2-microglobulin (B2M; 33,683 μg/mL), the anti-interleukin-6 receptor antibody tocilizumab (TCZ; 400 mg) was administered intravenously. One day after injection, he was afebrile. Four days after the TCZ injection, his CRP level dropped to 0.27 mg/dL, B2M level decreased to 3817 μg/mL, and viral load became low. No adverse drug reaction due to TCZ was observed. The patient was discharged 15 days after admission. The early administration of TCZ in this patient prevented the pneumonia and kidney injury caused by COVID-19 from progressing to hyperinflammation syndrome.

**Keywords:** COVID-19; pneumonia; kidney injury; tocilizumab; β2-microglobulin; CRP

## 1. Introduction

The epidemic of pneumonia first identified at the end of 2019 was caused by a new coronavirus named severe acute respiratory syndrome coronavirus 2 (SARS-CoV-2), which has phylogenetic similarity to SARS-CoV [1]. The disease triggered by SARS-CoV-2 infection, known as coronavirus 2019 (COVID-19), spread rapidly worldwide, causing varying degrees of illness. Patients with COVID-19 often present without fever, and many do not have abnormal radiologic findings [2]. Once SARS-CoV-2 infection is established, it can cause various symptoms, including fever, cough, sputum, sore throat, and pneumonia, in previously healthy people. While most infected individuals experience a mild course of illness, those who are middle-aged or older and those who have a pre-existing condition tend to develop more severe symptoms [3]. Even patients who appear to have only a mild pneumonia at the beginning of infection can proceed to develop acute respiratory syndrome (ARDS) and/or multiple

organ failure over the course of the illness [4]. Various drugs, including ciclesonide [5], nafamostat mesilate [6], and favipiravir [7] have been proposed to be effective against COVID-19.

Accumulating evidence suggests that a subgroup of patients with severe COVID-19 develop cytokine release syndrome (CRS; also known as cytokine storm syndrome), which is characterized by high circulating levels of pro-inflammatory cytokines that result in direct tissue injury, especially in the lungs. In cases of hyperinflammation, immunosuppression is likely to be beneficial [8]. The pro-inflammatory cytokine interleukin (IL)-6 plays a key role in CRS, and it has a prominent role in the inflammatory cascade [9,10]. Tocilizumab (TCZ) is a humanized monoclonal antibody that acts as an inhibitor of both membrane-bound and soluble IL-6 receptor. It has been tested as a remedy for CRS associated with COVID-19, and TCZ administration was reported to cause significant clinical improvement in COVID-19 patients with pneumonia requiring a ventilator [11,12]. However, some other studies did not confirm this effect [13,14]. Hyperinflammatory syndrome (HIS), which can be caused by COVID-19, is a systemic inflammatory syndrome associated with thromboembolic diseases, such as acute ischemic strokes caused by large vessel occlusion or myocardial infarction, encephalitis, acute kidney injury, and vasculitis (e.g., Kawasaki-like syndrome in children and renal vasculitis in adults) [15], and 40% of COVID-19 patients experience proteinuria and hematuria, which are suggestive of kidney infection and injury [16]. Recent work has proposed the evaluation of urinary biomarkers, such as β2-microglobulin (B2M), as potentially useful for identifying COVID-19 patients with active CRS who are likely to become critically ill [17].

The present report describes the case of a COVID-19 patient who showed respiratory insufficiency with elevated inflammatory and kidney injury markers and was treated with TCZ to determine if this drug could have a therapeutic effect on a patient with pneumonia and kidney injury.

## 2. Case Presentation Section

A middle-aged COVID-19 patient (49 years old, male) was admitted to our hospital on 22 June 2020. He presented with a five-day history of high fever, watery diarrhea, dyspnea, cough, and anosmia. A SARS-CoV-2 infection was confirmed by a positive SARS-CoV-2 PCR test result obtained from a nasopharyngeal swab from the patient. He had a history of hypertension and of smoking one pack of tobacco cigarettes per day. The patient was routinely taking the following drugs as anti-hypertensive agents: isoprolol fumarate (β-receptor blocker; 5 mg/day), olmesartan medoxomil (angiotensin II receptor blocker; 20 mg/day), doxazosin mesilate (adrenergic alpha1 antagonist; 1 mg/day), and amlodipine besilate (calcium channel blocker; 5 mg/day).

Laboratory findings for the patient included elevated levels of C-reactive protein (CRP; 1.35 mg/dL), lactate dehydrogenase (LDH; 250 U/L), ferritin (406 ng/mL), D-dimer (0.8 μg/mL), and urinary B2M (4960 mg/mL) (Table 1). A chest computed tomography (CT) scan showed multiple patchy ground-glass opacities (GGO) in both lobes of his lungs (Figure 1a). The patient's vital signs were as follows: heart rate of 108 beats/min, respiratory rate of 33 breath/min, axillary temperature of 39 °C, oxygen saturation (SpO$_2$) in ambient air of 88%, and blood pressure of 148/98 mmHg (Figure 2). Oxygen administration (2 L oxygen flow) was started in response to the hypoxemia. Azithromycin (500 mg/day), ciclesonide (200 μg inhaler; two inhalations per day), nafamostat mesilate (40 mg/day), and favipiravir (3600 mg on the first day, 1600 mg thereafter) were administered. Despite these treatments, the patient's clinical condition on day 3 was not improved. Specifically, his high fever persisted, his hypoxemia worsened, and 5 L of oxygen flow was needed to keep his SpO$_2$ above 94%. Additionally, the patient's levels of CRP, ferritin, D-dimer, and urinary B2M had increased (Figure 2, Table 1), and he complained of intrathoracic agony and severe cough. Chest CT findings from day 3 revealed a worsening of the GGOs (Figure 1b) and a reticular pattern in both lobes of the lungs.

This patient signed a case report consent form.

**Table 1.** Laboratory data from the patient during his hospitalization.

| Laboratory Data | Reference Range | Day 0 | Day 3 | Day 7 | Day 14 |
|---|---|---|---|---|---|
| **Blood test** | | | | | |
| White-cell count (/μL) | 3700~8500 | 4600 | 5400 | 7100 | 5200 |
| Neutrophils (%) | 44.0–68.0 | 71 | 73.6 | 69.1 | 62.9 |
| Bands (%) | 0.0–10.0 | 0 | 0 | 0 | 0 |
| Metamyelocytes (%) | 0 | 0 | 0 | 0 | 0 |
| Lymphocytes (%) | 27.0–44.0 | 22.7 | 17.9 | 20.1 | 29.8 |
| Monocytes (%) | 3.0–12.0 | 5.9 | 8.1 | 6.9 | 4.2 |
| Eosinophils (%) | 0.0–10.0 | 0.2 | 0.2 | 3.5 | 2.7 |
| Basophils (%) | 0.0–3.0 | 0.2 | 0.2 | 0.4 | 0.4 |
| Hematocrit (%) | 42.0–53.0 | 42.6 | 41.2 | 44.0 | 40.1 |
| Hemoglobin (g/dL) | 13.5–17.5 | 14.2 | 14.2 | 15.5 | 13.8 |
| Platelet count (/μL) | 150,000–355,000 | 157,000 | 179,000 | 304,000 | 271,000 |
| Red-cell count (/μL) | 3,900,000–5,300,000 | 4,670,000 | 4,560,000 | 4,860,000 | 4,400,000 |
| **Biochemical test** | | | | | |
| Urea nitrogen (mg/dL) | 2–80 | 12 | 13 | 21 | 17 |
| Creatinine (mg/dL) | 0.65–1.07 | 1.15 | 1.06 | 1.42 | 1.23 |
| ALT (U/L) | 3–40 | 25 | 29 | 53 | 69 |
| AST (U/L) | 8–35 | 24 | 35 | 54 | 33 |
| LDH (U/L) | 124–222 | 250 | 402 | 771 | 251 |
| Ferritin (ng/mL) | 14–304 | 406 | 559 | 735 | 694 |
| CRP (mg/dL) | 0.00–0.3 | 1.35 | 3.19 | 0.27 | 0.05 |
| Total protein (g/dL) | 6.6–8.4 | 7 | ND | 6.9 | 6.3 |
| Albumin (g/dL) | 3.8–5.2 | 4 | ND | 3.7 | 3.6 |
| **Coagulation test** | | | | | |
| PT (s) | 11.2 | 12 | 11.8 | 11.3 | ND |
| PT/INR | 70.0–110.0 | 97.3 | 95.0 | 104.6 | ND |
| APTT (s) | 23.0–38.0 | 30.3 | 30.5 | 22.7 | ND |
| D-dimer (μg/mL) | 0.00–1.00 | 0.85 | 1.18 | 1.25 | 1.12 |
| Fibrinogen (mg/dL) | 200–400 | 568 | ND | ND | ND |
| **Urine test** | | | | | |
| Color | Yellow | Yellow | ND | ND | ND |
| Clarity | Clear | Clear | ND | ND | ND |
| Specific gravity | 1.009–1.025 | 1.03 | ND | ND | ND |
| pH | 4.8–7.5 | 5.5 | ND | ND | ND |
| Protein | - | ++ | ND | ND | ND |
| sugar | - | - | ND | ND | ND |
| White cells per high-power field | - | - | ND | ND | ND |
| Red cells per high-power field | - | - | ND | ND | ND |
| β2-microglobulin (μg/L) | 30–340 | 4960 | 33683 | 3817 | 508 |
| **SARS-CoV-2 PCR** | | | | | |
| Viral load (copy/test) | 0 | 480.9773 | ND | 219.5791 | 1.9219 |

Abbreviations: ALT, alanine aminotransferase; AST, aspartate aminotransferase; LDH, lactate dehydrogenase; CRP, C-reactive protein; PT, prothrombin time; PT/INR, prothrombin time international normalized ratio; APTT, activated partial-thromboplastin time; -, normal; ND, not done; ++, moderate positive.

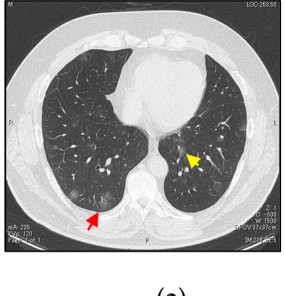 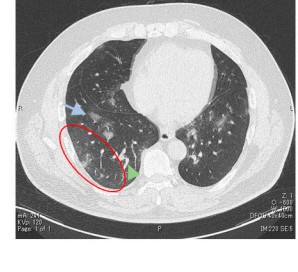 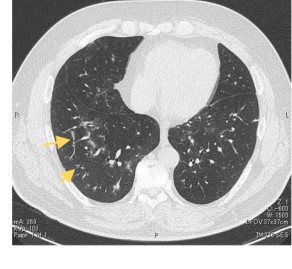

       (**a**)               (**b**)               (**c**)

**Figure 1.** Computed tomography (CT) findings of the patient's lungs at days 0, 3, and 14. (**a**–**c**) CT images of the patient's lungs at day 0 (**a**), day 3 (**b**), and day 14 (**c**). Red and yellow arrows in (**a**) indicate bilateral ground-glass opacities (GGOs) with a peripheral and posterior predominance, respectively. Red circle in (**b**) indicates an increased number of bilateral GGOs that are widespread in the lungs. Blue and green arrows in (**b**) indicate a reticular pattern and the appearance of dilated vessels, respectively. Orange arrows in (**c**) indicate GGOs that have shrunk, changed into a small consolidation, or developed a cord-like appearance.

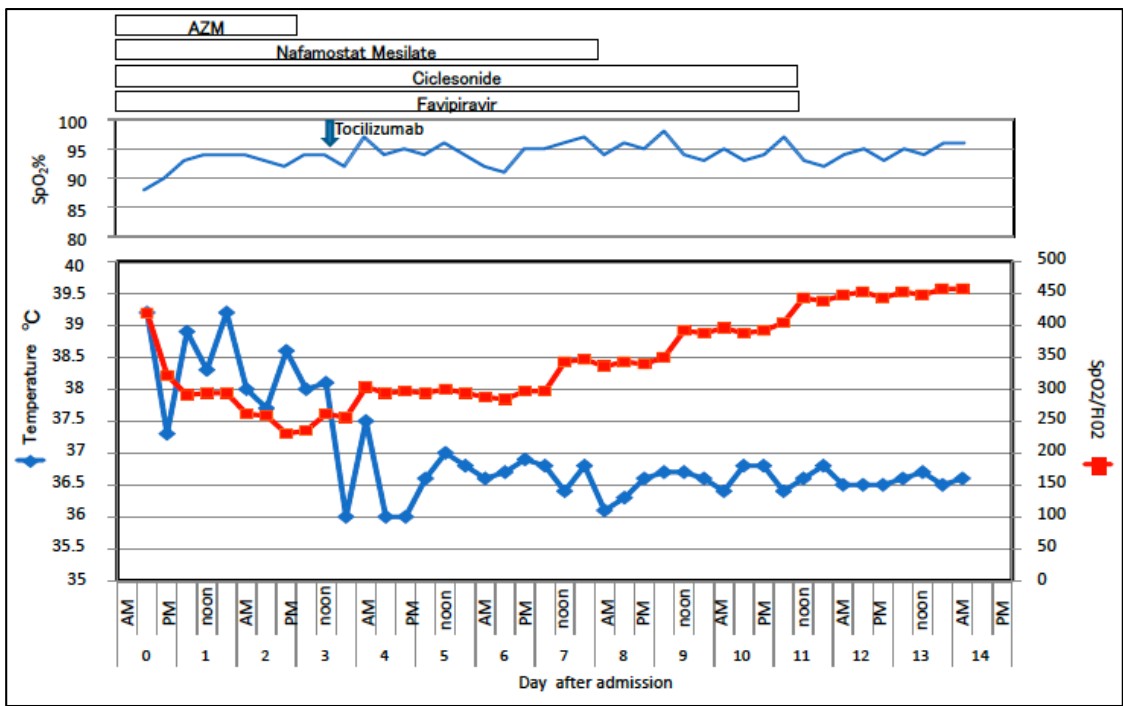

**Figure 2.** Fluctuations of the patient's oxygen saturation (SpO$_2$), body temperature, and SpO$_2$/FIO$_2$ ratio during his hospitalization. The drug dosing period is shown at the top of the figure.

Because we suspected that this patient had CRS, we administered TCZ (400 mg) intravenously with a histamine H1-receptor blocker on day 3. One day later (day 4), he became afebrile, and his other clinical symptoms, such as shortness of breath, cough, sputum, and general fatigue, were ameliorated within a few days. Four days after the TCZ injection (day 7), the patient's levels of CRP and urine B2M decreased, and his platelet count increased. Notably, the patient's levels of LDH, ferritin, and D-dimer all temporarily increased at 4 days after TCZ administration (day 7), followed by declines. Because the patient's condition had improved, the nafamostat was discontinued on day 7, and the ciclesonide and favipiravir were discontinued on day 11 (Figure 2). On day 11, the patient's SpO$_2$ in ambient air increased to 95%. Eleven days after the TCZ injection (day 14), the patient's laboratory findings improved (Table 1). A chest CT scan at that time revealed that the GGOs had reduced in size and that the lesions had changed into a small consolidation or developed a cord-like appearance (Figure 1c). The viral load of SARS-CoV-2 as measured by PCR also decreased (Table 1). There were no significant adverse effects observed after the administration of TCZ, except for a slight increase in the levels of some coagulation markers, such as D-dimer and ferritin, as described above. Fifteen days after admission, the patient was discharged.

## 3. Discussion

The patient in this case presented with a five-day history of high fever, watery diarrhea, dyspnea, cough, and anosmia. After admission, his symptoms worsened. He was treated with three different drugs that have been reported as effective against COVID-19 (ciclesonide [5], nafamostat mesilate [6], and favipiravir [7]), but no beneficial effects from these treatments were observed. A worsening of the patient's pneumonia, accompanied by increases in his levels of CRP, ferritin, D-dimer, and urinary B2M, led us to suspect that he was developing HIS/CRS.

The severity and outcome of COVID-19 appear to be related to the degree and characteristics of the immune response to SARS-CoV-2 infection. Patients who developed acute respiratory distress syndrome are believed to suffer from HIS/CRS with circulating pro-inflammatory cytokines. Because IL-6 levels are elevated in HIS/CRS during severe cases of COVID-19, the monoclonal anti-IL-6 antibody

TCZ is recommended for treatment [10]. In this case, a single dose of 400 mg was given intravenously, as previously described [13]. COVID-19 pneumonia with ARDS has been shown to be characterized by HIS. In Italy, TCZ treatment was found to be associated with rapid, sustained, and significant clinical improvement [11]. Preliminary data collected in China show that TCZ treatment, which improved the clinical outcome immediately in severe and critical COVID-19 patients, is effective at reducing mortality [12]. However, another report from Italy demonstrated that the clinical improvement and mortality were not statistically different between patients treated with TCZ or the standard treatment at day 28. Additionally, bacterial or fungal infections were recorded in 13% of the TCZ patients and in 12% of the standard treatment patients in that study [13]. Furthermore, TCZ administration was not found to reduce the intensive care unit admission or mortality rates in a cohort of 21 patients, thus it was proposed that additional data are needed to understand the effect(s) of TCZ in treating patients diagnosed with COVID-19 [14].

Treatment for HIS/CRS is more likely to be successful when it is commenced early in the disease course. In our case, the patient's levels of CRP, ferritin, and B2M levels were elevated (Table 1), and he had a high fever and hypoxemia (Figure 2), which are suggestive of HIS. Because CRP is synthesized by IL-6 [18], and the serum levels of both IL-6 and CRP are significantly correlated with COVID-19 severity [19], we administered TCZ to the patient. Fortunately, the administration of TCZ was extremely effective. TCZ binds to the IL-6 receptor, consequently inhibiting the IL-6 signal. Although suppressing the IL-6 signal has the potential to exacerbate an infection, in this case, the viral load decreased anyway. The temporary rise in LDH, D-dimer, and ferritin levels after TCZ administration on day 7 may indicate persistent hypercoagulability. A dramatic elevation of the D-dimer level on day 5 was also reported; thus, the risk of thrombotic complications after TCZ treatment might not have been completely reduced [20]. One patient with COVID-19 complicated by CRS who was treated with TCZ progressed to secondary hemophagocytic lymphohistiocytosis, and another developed viral myocarditis. These cases indicate a need for clinical trials to determine optimal patient selection and timing for the use of TCZ in treating COVID-19 [21]. It should be noted that the good prognosis observed in our case is not necessarily the result of only the TCZ treatment, as this patient received a variety of therapeutic agents. The possibility of spontaneous recovery should also be considered.

Although melatonin therapy has been advocated as a more economical option for treating HIC/CRS in COVID-19 infection, melatonin is not currently available for use in medical therapy in our country [22]. In SARS-CoV-1, a viral protein encoded by ORF8b directly interacts with inflammasome nucleotide-binding domain leucine-rich repeat (NLR) and pyrin domain-containing receptor 3 (NLRP3). Melatonin was reported as an inhibitor of NLRP3 inflammasome [23]. Accordingly, a clinical trial assessing the use of an intravenous perfusion of an injectable melatonin formulation in ICU patients suffering from COVID-19 is planned [24].

In our case, B2M was also good marker for disease severity and response to therapy. Evaluating urinary biomarkers such as B2M may help to identify COVID-19 patients with active CRS who are likely to become critically ill [17]. Acute kidney injury is largely associated with hemodynamic instability [25]. B2M is freely filtered by the glomerulus and completely reabsorbed by proximal tubular cells. The impaired uptake of B2M as a result of tubular injury results in increased B2M urinary excretion; thus, B2M can be considered a direct marker of tubular dysfunction [26]. Therefore, B2M levels should be monitored in COVID-19 patients. Because it has been previously reported that the IL-6 level does not decrease significantly in the short term after treatment with TCZ [12], a better biomarker for assessing the therapeutic effect of TCZ should be developed. Recently, we reported that the plasma osteopontin (OPN) levels can reflect acute kidney injury in leptospirosis [27]. Notably, IL-1β-induced IL-6 and sIL-6R caused an upregulation of OPN in THP-1 macrophages [28]. Therefore, it is possible that plasma OPN levels might be a good marker for TCZ therapy.

There is concern that the use of an angiotensin II receptor blocker, which is a type of oral medication for hypertension, may be an exacerbating factor for COVID-19, because ACE2 is involved in the mechanism of SARS-CoV-2 infection. However, the American Heart Association recommends that

patients taking ACE-inhibitors or angiotensin II receptor blockers who contract COVID-19 continue their treatment [29,30]. In the present case, the patient was taking four anti-hypertensive drugs, including an angiotensin II receptor blocker, prior to contracting COVID-19, and these treatments were not discontinued.

Despite Japan's early exposure to the COVID-19 pandemic, its dense and aging population, and its limited social distancing measures, this country has reported low rates of SARS-CoV-2 infection and death from COVID-19 [31]. It was suggested that potential differences in ACE2 expression among the different populations and ethnicities in Asia may cause a less efficient replication of SARS-CoV-2 [32]. Alternately, Japanese individuals may have unknown protective mechanisms against COVID-19, and such mechanisms could be responsible for the good outcome in this patient [31,33].

## 4. Conclusions

A 49-year-old COVID-19 patient with pneumonia and kidney injury was successfully treated with TCZ. A single intravenous administration of TCZ led to a remarkable recovery of the patient. This result suggests that early intervention using TCZ is effective against HIS/CRS development in COVID-19 infection.

**Author Contributions:** Conceptualization, Y.A. and T.H.; methodology, Y.A.; formal analysis, H.C.-Y.; resources and data curation, M.H., Y.S., Y.K., and H.K.; writing—original draft preparation, Y.A.; writing—review and editing, C.Y.H. and T.H. All authors have read and agreed to the published version of the manuscript.

**Funding:** This research was funded by the Japan Society for the Promotion of Science (JSPS) Grants-in-Aid for Scientific Research (KAKENHI) (grant no. JP17H01690).

**Conflicts of Interest:** The authors declare no conflict of interest. The funders had no role in the design of the study; in the collection, analyses, or interpretation of data; in the writing of the manuscript, or in the decision to publish the results.

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
