# Peer review of "Successful Treatment of a COVID-19 Case with Pneumonia and Renal Injury Using Tocilizumab"

_reports, doi:10.3390/reports3040029_

Round 1

Reviewer 1 Report

Dear Author,

I read carefully your paper. Although it is appropriate to disseminate and share all information about the correct management of patients with COVID-19, even those derived from a single case, I believe that your case report is not suitable for publication, mainly for two reasons.

The first reason is that very few elements of novelty are descibed in your paper: the case described has no special pecularities that may contribute to the general knowlegde about COVID-19 managment.

The second, and most important reason is that you do not provide any conclusive evidence about the actual link between tolicizumab administration and the patient's clinical improvement, other than a mere temporal link. Tolicizumab was administered on the basis of clinical worsening, only. Evidence of the beneficial effect of tolicizumab could have been provided by the pre- and post-dose IL-6 dosing, but these dosages were not performed. According to the description of the case, the improvement of the patient could be due to effects of other drugs, or being caused by the clinical course of disease, that spontaneusly improve after few days in most cases.

Moreover, many minor mistakes are present:

  • the introduction is too long, and some sentence (such as "We are hesitant to administer the drug because of these side effects.") should be included in the discussion,istead;
  • the correct Tociluzumab dosing is decidedon the basis of the body weight, that is not reported in the paper. Thus, I'm not able to understand if 400mg is the correct drug dose;
  • viral load is not affected by tocilizumab, so the speculation about a link between viral load decrease and tocilizumab administration is not appropriate;
  • The d-dimer increase, that you suggest to be due to tolicizumab light side-effect, is on the contrary well described as being part of clinical course of COVID-19;
  • The English language needs some revisions.

Reviewer 2 Report

Comments:

The general lines, English should be improved throughout the paper. Authors should avoid passive or impersonal phrases (line 43, line 61,...)

Introduction

Line 48-49: tis sentence is not well understood.

Line 52: in the introduction, authors should avoid speaking in the first person and should avoid talking about your study

Line 69-70: the conclusion must be better expressed. Authors should better express what their objective was.

Case presentation section

Figure 1: Authors should better explain the difference between the images and point it out in the figure

Line 95: the title of the table is not correct

In the table 1, Differential ?

Discussion

The discussion section should begin with the main finding of the study

Line 118, 120, 140,143,...: authors should avoid passive or impersonal phrases.

Authors Contribution style is not correct

References style is correct.
